# The Lifetime History of the First Italian Public Extra-Corporeal Shock Wave Lithotripsy (ESWL) Lithotripter as a Mirror of the Evolution of Endourology over the Last Decade

**DOI:** 10.3390/ijerph20054127

**Published:** 2023-02-25

**Authors:** Rafaela Malinaric, Guglielmo Mantica, Mariano Martini, Federica Balzarini, Federico Mariano, Giovanni Marchi, Piero Tognoni, Daniele Panarello, Paolo Bottino, Carlo Terrone

**Affiliations:** 1IRCCS Ospedale Policlinico San Martino, 16132 Genova, Italy; 2Department of Surgical and Diagnostic Integrated Sciences (DISC), University of Genova, 16132 Genova, Italy; 3Department of Health Sciences, University of Genoa, 16132 Genova, Italy

**Keywords:** ESWL, endourology, renal stones, urolithiasis, shockwave

## Abstract

Extracorporeal shockwave lithotripsy (ESWL) is the only non-invasive treatment for kidney stones. It does not require an operating room, anesthesia, or hospital stay. Its role evolved over the years and nowadays ESWL is slowly disappearing from many stone centers and urologic departments. We present the history and the role of ESWL treatment since its birth in 1959 and its development through the following years. We also present details of its application and impact on the first Italian stone center in 1985. ESWL has had different roles over the centuries: in the early years it was a great alternative to open surgery and percutaneous nephrolithotripsy (PCNL), then it had its decline with the introduction of the miniscopes. Currently, although ESWL is not considered a treatment of excellence, newer models are emerging. With the application of new technologies and artificial intelligence, this technique can become a good option alongside endourologic treatments.

## 1. Introduction

Urolithiasis is one of the most frequent urological diseases, with prevalence from 1% to 20%, depending on geographic, dietary, genetic, and climatic factors [1]. It is far more common in arid and hot climates, as well as in men than in women, although in the past few years, the rates between sexes have almost equalized. Contrary to past opinions, urolithiasis is now considered a systemic disease; its incidence and risk factors have been correlated with diabetes, obesity, hypertension, metabolic disease, cardiovascular disease, and chronic kidney disease. Moreover, the quality of life in stone formers is worse than in non-stone formers [1,2].

The treatment of kidney stones depends on various factors, and currently, we have a lot of techniques and approaches available at our disposal, from the most invasive open surgery to the non-invasive extracorporeal shock wave lithotripsy (ESWL) [3,4,5,6] (Figure 1).

ESWL is not only non-invasive but is also the only treatment that does not require an operating room, hospital stay, and anesthesia. The procedures involved are described below.

During ESWL, externally generated shock waves are transmitted through the bodily tissues and concentrated onto the stone. Shock waves act in two different ways, by exerting positive pressure during the compressive phase and negative pressure during the tensile phase, causing stone fragmentation. Shear stress, compression, and spallation are the main physical phenomena that occur during extracorporeal shock wave lithotripsy. Furthermore, there are three principal shock wave generators: piezoelectric, electromagnetic, and electro-hydraulic [7].

The main indications for extracorporeal treatment are kidney stones that are not bigger than 15 mm, have a density lower than 800 HU, and skin-to-stone distance of 12 cm or less. The patient’s habitat and anatomy need to be taken into consideration before proposing it, as well as comorbidities such as coagulopathy or renal artery aneurysms [1].

Compared to percutaneous nephrolithotomy (PNL) and ureteroscopy (URS), there are fewer overall complications with SWL, with about 1% requiring hospitalization, such as renal symptomatic hematoma or nephrectomy due to a renal rupture (there are just a few cases in the literature).

Currently, the ESWL is slowly disappearing from many stone centers and departments and many endourologists are no longer comfortable with this procedure, despite the fact that it is a cheaper and safe technique. This is probably because of its large use in the years to the detriment of patients selection and as a direct consequence more re-operations.

Furthermore, year after year, ESWL is becoming overshadowed by brand-new surgical techniques and technologies, which require the use of operating rooms and anesthesia.

In this light, the aim of this review is to provide a complete overview of the history of ESWL from its birth to the present day, in particular by analyzing the history of the first Italian public lithotripter.

## 2. A Brief History of an ESWL Lithotripter

Determining when and where extracorporeal lithotripsy was born is not a simple task. In 1950, Russian engineer Lew Alexandrovitch Yutkin patented the principle of shock waves in the disintegration of kidney stones using an endoscopic electro-hydraulic generator. He developed this idea after observing how a lightning bolt could smash a log under water and applying the same principle to the plates.

The first medical application of this technique was achieved in 1959 by Victor Goldberg (Head of the Urological Department of the Municipal Hospital in Riga (Republic of Latvia)). According to Yutkin’s instructions, the engineer Leo W. Rese created an impulse generator and an electric stone probe, which enabled Victor Goldberg to successfully perform the first electro-hydraulic lithotripsy [3].

This device, composed of a condenser rack and bipolar electrodes, was efficaciously used during the early 1970s. One year later, the idea of contactless stone fragmentation using continuous ultrasound started to entertain the minds of enthusiastic scientists. Unfortunately, the damage induced when applied to biological tissues was too great to continue this project. However, the concept of contact-free fragmentation was never completely abandoned [8,9].

At Dornier Institute, Germany, during the late 1960s and early 1970s, the effect of shockwaves on solid materials was intensively studied. At that time, the Dornier Institute was an aircraft manufacturer concerned with supersonic aircraft dents and erosions caused by rain and micrometeorites.

In 1966, at the Dornier company, the effect of shockwaves on human tissues was discovered by accident. During experiments with high-velocity projectiles, one employee touched the plate at the same moment that the projectile hit the plate. He felt something in his body something like “an electrical shock”, but the measurements showed that no electricity was present. The generated shockwave traveled from the plate, through his hand, and into his body [10].

This is how the effect of shockwaves on biological tissues was discovered, and the first tests on animal tissues were launched [9,11].

The first experimental shockwaves, concentrated on kidney stones, were generated by underwater spark discharge, which subsequently traveled through a semi-ellipsoid [10,12].

From 1968 until 1971, the interaction between shockwaves and biological tissue in animals was investigated with a research program in Germany supported by the German Department of Defense [5].

The following investigations and cooperation together between engineers and physicians lead to a mutual objective—disintegrating kidney stones using shockwaves. In the beginning, the technical and medical insight of the idea was not well defined, but it was gradually developed and implemented. In 1971, Haeusler and Kiefer reported the first in-vitro disintegration of a kidney stone using shockwaves without any direct contact with the stone. Further in-vitro experiments of contact-free stone disintegration followed [10].

In 1974, the physicists at the Dornier Institute demonstrated the successful disintegration of kidney stones. The Department of Research and Science of Germany supported the research program called “Application of the ESWL”, and the team of scholars and German urologists formed included giants such as Eisenberger, Chaussy, Brendel, Schmiedt, Forßmann, and Hepp. They started with in vitro, and later in vivo, experimental phases.

Back then, there was not any technology that could adequately measure the expansion of shock waves, which led to the development of piezoelectric pressure probes. Furthermore, whilst the first prototype had a rubber membrane that did not allow optimal shock wave transmission, a water bath was integrated into the second. This, and some other minor adjustments were necessary to give life to the first official lithotripter—TM1. Animal experiments began in 1975 and were led by the same group of urologists. They tested the efficacy of the machine by placing human kidney stones in dogs’ calyceal systems [11,12,13,14]. Later that same year, a TM2 lithotripter with an integrated ultrasound scanner was built [9,11]. Unfortunately, fragmenting stones using ultrasound was not shown to be not particularly successful, which was the reason that, in 1978, the X-ray system was installed [11,13]. In no time at all, after proving that 3D localization of kidney stones with two-plane optical systems using x-rays was possible, the first Human Model Lithotripter (HM1) [13,15] was constructed and placed in Ludwig-Maximilians University, Klinikum Grosshadern, Munich. In order to produce smaller fragments, the shockwave energy was decreased while increasing the emission frequency. Working with these power/frequency settings, the possibility of kidney damage diminished as well. Interestingly, after the lithotripter was ready to be used in everyday practice, technical developers refused to take the responsibility for the patients’ security, so all the further clinical applications depended solely on this team of courageous doctors. Furthermore, to improve their skillset and expertise, intensive training in kidney stone localization using HM1 on volunteers started [9,11].

1980 was the most exciting year in the revolution of kidney stone treatment. On 7 February, the first patient was treated using the HM1 extracorporeal lithotripter. He successfully passed all the fragments, having no procedure-related complications. However, another unanticipated problem emerged—the shock wave induced extrasystoles. This inconvenience was rapidly addressed by the EKG triggering of the shock wave impulse, although the phenomenon lacks an explanation to date [9,11]. That same year, the first studies on this novel technique were published; however, the technique itself was not yet widely accepted. As a matter of fact, the American Urological Association refused to accept a submitted presentation regarding this technique. It was the next year that was crucial for welcoming this modern, non-invasive kidney stone treatment. Some big milestones were achieved, beginning with the name attributed to the procedure—‘Extracorporeal Shockwave Lithotripsy’, or ESWL, today a registered trademark of Dornier MedTech Systems, followed by establishing the main contraindications, valid to date. Initially, the procedure required anesthesia, with drugs administered via the intrathecal route but was substituted shortly afterward by the peridural. When the procedure was demonstrated to be safe and feasible, the HM2 lithotripter with improved optical and ergonomic systems was constructed. Finally, after almost a decade of hard work and dedication, in 1982, the first world’s lithotripsy center was established in Munich, Germany.

On the other hand, the first US studies began in the early 1980s, and Extracorporeal shockwave lithotripsy (ESWL) as a completely noninvasive therapy to break up stones within the kidney and ureter was introduced in 1983. Following the Food and Drug Administration (FDA) approval, the first lithotripter was installed and operative in 1984 in Indianapolis [9,11].

The Department of Urology at the University of Florida was one of six sites within the United States to investigate the efficacy of ESWL led by Dr. Birdwell Finlayson [16].

Finlayson was a world-renowned expert in stone disease. He was a clever researcher and clinician whose pioneering work in the field of urolithiasis led him to worldwide prominence in urology and was one of the six original co-investigators for shock wave lithotripsy in the United States [17].

In 1985, the first clinical treatment of a gallbladder stones with ESWL was performed in Munich (Germany), and in 1986, a prototype of a lithotripter without a bathtub was tested in Mainz [10].

By the end of 1985, already two hundred HM3 lithotripters were operative all around the world. Fun fact: the famous ‘C-arm’ fluoroscopy system, one of the modern lithotripter’s main features, was ideated and developed by the Israeli medical manufacturer, Direx Medical Systems Ltd., in the late 1980s [8,9].

## 3. The Extracorporeal Shock Wave Lithotripter’s Arrival to Genoa and Construction of the First Italian ESWL Center

In October 1985, the first national public lithotripter arrived in Genoa, only one year after its FDA approval. The introduction of this novel, non-invasive technology radically changed the management of renal and ureteral stones in the Italian metropolis. From 1970 up to the moment of the lithotripter’s arrival, more than 3000 open surgeries for kidney stones were performed at our department, and for that precise reason, it was very welcomed by the urologists at our Clinic. To adequately accommodate it, Genoa’s ‘Kidney Stone Center” (Centro Calcolosi) was purposely built a few months earlier. Although ideated and projected by the very best Italian architects and engineers, its construction was overseen and carefully guided by our urologists dedicated to urolithiasis, especially Dr. Bottino, who would later become the head of the section for the next 30 years. He introduced some minor modifications regarding the height and diameter of the walls surrounding the lithotripter to gain more sound isolation, as well as improved space-oriented distribution.

The enthusiasm was growing among the patients too, so much so that by March 1986, 340 of them had already been treated by the Dornier HM3. It quickly became the most used treatment for urolithiasis, even when compared to surgical procedures such as ureterorenoscopy (44 pts) and percutaneous nephrolithotripsy (220 pts) during the same period (Figure 2). At that time, endourology could not replace open surgery because of a lack of radiological equipment. On the other hand, the lithotripter enabled the treatment of many more patients and almost replaced PCNL and open surgery. Only two years after using it, PCNL and OS were given well-defined indications and were reintroduced in our everyday practice.

As for the stone position, size, and composition, the majority of ESWL-treated uroliths were calcium oxalate type, caliceal, and around 10 mm. Furthermore, all patients underwent a follow-up with abdominal x-rays after being treated in order to evaluate the efficacy of the treatment itself. Generally, 79% of all the patients were declared stone-free within 3 months, and the percentage rose to 85% if the stone was ureteric. We need to take into consideration that this is the data from the very beginning of the ESWL use at our center, and stone-free rates do not differ a lot from those reported today. Months passed and the passion for the Dornier machine was growing more, and more, and more. Our equipment team, led by Dr. Puppo, even proposed an index for the ESWL stone treatment (stone volume x urinary tract infection/compliance of urinary tract x patient cooperation) [14,18].

## 4. The Lithotripter’s Golden Ages

The Lithotripter’s Golden Era started in 1989 and lasted until 2004 when we treated at least 800 patients annually [15,19]. This was possible thanks to the introduction of piezoelectric lithotripters that offered the possibility of less painful treatments. There were also some minor adjustments performed on the Dornier HMIII, such as the installation of the new generator and enlargement of the semi-ellipsoid. This increased the area of shock wave entry and consequently reduced the pressure on the area treated, which helped in reducing the pain.

In the operating room, a combination of treatments became more and more frequent; for example, staghorn calculi were treated with percutaneous nephrolithotripsy and extracorporeal lithotripsy in tandem. This debulking technique provided the positioning of the ureteral catheter, accessing the renal pelvis from the inferior calyx first, starting the PCNL, liberating the pyelo-ureteral junction, and balloon-dilation of the renal cavities to favor the passage of residual fragments. Major residual fragments were subsequently treated by an extracorporeal approach [16,20].

In that period, with the new lithotripters, anesthesia was a thing of the past, and ESWL became an outpatient treatment. Beforehand, the hospital stay was around 2.8 days [15,19]. Prior to the treatment, the only medication used was promazine for sedation, and metamizol for analgesia. The recommendations given to the patients were the following: control abdominal x-ray on the first and third day after the treatment (before the consultation with the referred urologist), and oral administration of antibiotics and analgesia only if needed. Complications requiring hospitalization were reported in 1% of patients, and we witnessed huge benefits in doing it as an outpatient procedure: the costs of the kidney stone treatment lowered greatly, and patients were more content avoiding the operating room. Certainly, there were some calculi in positions difficult to treat, such as stones placed in anterior calyces of ectopic kidneys and in the iliac ureter; nevertheless, the requests for the ESWL kept growing. Consequently, many suggestions were made on how to treat them, and it was our team, in January 1987, that proposed the position later accepted worldwide—ESWL with patients in the prone position. This position lowers the exposition to radiation, making it more secure for the patients, and currently, it is the preferred one [17,18,21,22].

The excitement around ESWL became even more evident in the late 1980s and spread all around Italy.

The machine that provided an outpatient, painless treatment of renal calculi, significantly lowered hospital costs and avoided major surgery-related risks to the patient? Everybody wanted a piece of it! Companies wanted to produce it, urologists wanted to work with it, and patients wanted to be treated by it.

There were still manufacturers that could not produce the machine that executed at levels that Dornier HMIII did, even six years after the beginning of its clinical application! Then, in 1989, Genoa welcomed Lithoring Multi-One, a third-generation Italian lithotripter (Figure 3). Lithoring Multi-One had a challenging task—to perform as well as Dornier HMIII—which set very high standards. Now, our team started to test the national product and the first report about its performance was published in 1990 [18,22]. Our urologists were chosen to test it because of the experience they had accumulated with the Dornier’s lithotripter, and the fame they had gained all over the country. This was not an isolated case, they were asked to test almost all the innovative devices in the field of endourology as well, such as the first flexible ureterorenoscope and more [19,23]. Our ‘Kidney-stone’ center became the national referral center, and urologists such as Dr. Puppo and Dr. Bottino, became nationally acknowledged experts.

The Lithoring lithotripter had an integrated radiological table, C-arm fluoroscopy, ultrasound, and shockwave systems. It was really a modern and avant-garde machine. All health-care professionals were impressed by its functionality and design, which permitted the alignment of the lithotripter with the center of the fluoroscopy beam. Furthermore, the bidimensional position of the calculus was taken by the 30° degree C-arm rotation, and it offered the ability to switch from fluoroscopy to ultrasound imaging without moving the patient, reducing the exposure to the radiation of both the physician and the patient. This was incredible progress when compared to the first prototype created only nine years prior. The reported stone-free rate was 83.3%, whereas the retreatment rate was as low as 1.16% for pyelic stones, and 1.53% for ureteral stones, slightly higher [15,19].

Our ‘Kidney-stone’ unit was flourishing, attracting patients even from outside of the Liguria region. In the early 1990s, 1992 and 1993 to be precise, the number of patients treated was circa 1050 annually, reaching its peak. The hospital administration provided us with a small operating room, radiological unit, radiologist, specifically trained nurses, technicians, and secretary that were available at all working hours. If required, the patients had practically all-day free access to the unit, without the need of making an appointment or accessing the ER. Obviously, if clinical symptoms indicated worsening of the patients’ state, they were referred to the intensive care unit. The initial vision of urologists dedicating their lives to the center and its patients, and just maybe utopia, was eventually realized. 

Then, in 1994, the number of patients undergoing ESWL started to fall.

## 5. The Rise of URS/RIRS and the Downfall of Lithotripter—Myth or Reality?

In 1990, one of the first papers describing the two-year-long experience using the flexible ureterorenoscope by Dr. Puppo was published. The idea to construct such a device dates back to the 1970s, and its use was already reported by Takayasu and Aso [20,24]. This device permitted the treatment of calyceal stones and exploration of all the renal collecting system without the need of performing open surgery. Before being tested in Genoa, this surgical technique was evaluated by Bagley, Streemand Aso, in the late 1980s [21,22,23,25,26,27]. Indications for performing it were undiagnosed hematuria, evaluation of filling defects, and treatment of residual fragments in the renal calyces after PCNL and/or ESWL. The first device used was the 10.8 Fr ureterorenoscope manufactured by Olympus that offered 160–80° deflection and a 1.2 mm working channel. Our team reported the same operative difficulties that we encounter today: low visibility during hematuria, difficulty entering the ureter if the ureteral meatus was narrowed or exploring the calyx with a tight infundibulum. Although the instrument had a wider diameter and worse visibility than the ones currently in use, its great potential was immediately recognized. From 1994 to 2004 the number of patients treated with ESWL continued to fall but remained above 800 annually. In the same period, we registered a rise in the endoscopic surgical approach, using both, rigid and flexible ureterorenoscopes. It was slow and steady progress, going from 100 patients treated in 1994, to 180 in 2004. When the comparison by years was made, the number of patients treated for urolithiasis in toto was the same; however, the indications changed. The 1990s brought significant novelties in kidney stone treatment such as the development of the smaller rigid ureterorenoscopes, by substituting the rigid lenses with fiber optics, increasing the diameter of the working channel to 3 Fr, and improvement of the flexible ones, at the time named ‘miniscopes’ [24,28]. This made endoscopy of the upper urinary tract practically as minimally invasive as ESWL. After their introduction to the clinical practice, they were preferred for the treatment of the ureteral calculi to the ESWL, leaving extracorporeal treatment only for selected patients. Reported stone-free rates for pelvic ureteral calculi were significantly higher when compared to those of the ESWL, amounting to 93.6%. In that period there were no randomized control trials that compared these techniques, but Anderson et al. [25,29] published a paper stating that 96% of the patients with pelvic ureteral urolithiasis were stone-free when treated with Dornier HMIII, 83% when treated with Siemens Lithostar, and 100% when they underwent ureteroscopy. Furthermore, because of this significant upgrade, operative time was significantly lowered, hospital stay was shortened, and, consequently, endoscopic surgery became cheaper than ESWL outpatient treatments (the cost of overall ureterorenoscopy/endoscopic management of the ureteral calcoli resulted circa USD 8.263, whilst that of the ESWL HMIII circa USD 8.539, with all the ancillary maneuvers included) [25,29]. Moreover, if they underwent URS, patients were stone-free in shorter periods and retreatment rates were a lot lower. The so-called ‘miniscopes’ on the market were produced by various companies; however, the most used were Olympus’s, Wolf’s and Circon ACMI’s [26,30]. Papers published during the mid-1990s more and more frequently started to report all the listed above, and that marked the start of the lithotripter’s downfall. Reviewing our statistics, we noted a major and constant fall in the use of the lithotripter from 2004, with a 50% decrease in the number of patients treated in only 10 years, going from 800 to 400 yearly. This trend was accompanied by a rise in endoscopy treatments, with a 100% increase in the number of patients undergoing ureterorenoscopy in the same time period (Figure 4).

ECIRS (endoscopic combined intrarenal surgery) deserves to be mentioned: a recent, emerging approach for the treatment of large complex urolithiasis. It combines the anterograde and retrograde approach to the kidney; ECIRS allows the combined use of all the rigid and flexible endourological armamentarium, optimal visualization during percutaneous renal puncture, and final visual control of the stone-free condition.

## 6. Where Is Our Lithotripter Today and Where Is It Going to Be Tomorrow?

From 2014 to 2018, the number of patients treated with ESWL continued to fall, and the number of the ureterorenoscopies rose. In 2018, we performed circa 100 extracorporeal treatments and 330 endoscopies, while the number of PCNLs remained relatively constant throughout the 1980s, 1990s, and 2000s. Ergo, in the last four years we have seen an ulterior decrease of 125% in the number of patients treated with ESWL, and a 50% increase in endoscopic surgery. In addition, looking at the numbers in toto, there has been a general decrease in patients actively treated with all three techniques, going from 1125 in 1993 to 550 in 2018. This could be due to the introduction of the ‘active surveillance’ concept in the field of urolithiasis as well, already present in other urological diseases. Even though extracorporeal treatment does not require an operating room or anesthesia, and is considered non-invasive, shock waves do cause kidney damage, although minimal (loss of 2% of glomeruli per treatment) [27,31]. Unfortunately, our lithotripter stopped functioning in late 2018, and we have not offered any more of this kind of intervention. COVID-19 has greatly blocked investments dedicated to endourology. However, from 2023 the recently restored lithotripter will return to function, helping us to reduce the impressive waiting list accumulated in recent years of the pandemic, in particular for the treatment of kidney stones [32].

In fact, in recent years, we have offered surgery to everyone on the waiting list. However, we have several patients on the waiting list that require active treatment and could undergo both endoscopy and ESWL, and an outpatient solution could become important again, especially in the post-COVID-19 era.

Obviously, as with many other Institutions after the pandemic, we do not have that many hospital beds, operating rooms, or staff at our disposal. This causes patients to wait a lot, and some of them could develop more kidney damage or go into urosepsis while waiting. Other important problems are surgery and general/peridural anesthesia-related risks, that could be completely avoided if we treated patients in ambulatory settings. Further, ureterorenoscopy-related complications, such as urosepsis, are not uncommon, while other rarer events, such as complete ureteral avulsion, are disastrous.

One of the major pitfalls of the ESWL is a lack of standardization when used. We still do not have well-defined recommendations about the number of shock waves and maximum power to use during each session, nor whether the patients should be stented beforehand or not. Chaussy and Tiselius recently reviewed the literature regarding this, and concluded that lower rates of shock waves (60–90) result in better outcomes [28,33], as well as using ramping energy rather than fixed and pre-stenting the patients, but all these are merely the suggestions based on the reviewed articles.

We know that clear indications for use of the ESWL are kidney stones measuring 10–20 mm, that do not present density over 800 HU, and do not distance over 12 mm from the skin surface. Obviously, the patient’s anatomy and stone position need to be taken into consideration. After evaluating all the above, what kind of treatment the patient requires should be discussed with him, explaining the morbidity and stone-free rates of both procedures, ESWL and ureteroscopy.

Now, do we think that EWSL has no place as a treatment in the future? Absolutely not. On the contrary, we think it is still one of the essential tools urologists can use. However, the patients need to be carefully selected, the treatment individually tailored, and the machines used up to date, along with all the other technologies in urologic surgery. Undisturbed shock wave transmission from its source to the stone, the use of fewer shock waves, less power of shock waves to achieve stone disintegration, and lower complication rates are all the principal objectives of the new-generation lithotripters [28,33].

Firstly, after eliminating the water tub, in order to achieve optimal coupling and stone disintegration, there should not be any presence of air bubbles in the transmission medium. Therefore, the novel devices should have an integrateliked video camera of the transmission zone. Moreover, recent studies [29,34] demonstrated fragmentation of ureteral and kidney stones with 25% fewer shock waves by using a camera during the treatment. Secondly, by amplifying the focal zone, it is possible to eliminate the second, compressive wave. In 2013, a group of researchers added an annular ring to the lithotripter’s lens, documenting a 30% increase in stone fragmentation compared to the unmodified lens [35]. Another interesting concept is ‘burst wave lithotripsy’, a method where bursts of high-frequency ultrasound and ultrasonic propulsion are used to break up the stones. The ultrasound waves travel at a frequency of 200 Hz, meaning they were circa 150 times faster than those of the conventional ESWL [36]. By using a probe that relies on ultrasound, the peak pressures and cavitation bubbles formation decrease significantly, alongside radiation exposure and pain during treatment. Various techniques have been proposed in recent years to eliminate cavitation bubbles, such as combining ESWL and histotripsy, and the introduction of an integrated piezoelectric transducer. All the proposed technical modifications and adjunctives are still to become the gold standard and recommended by the guidelines.

Recently, we have witnessed ESWL turn to artificial intelligence for decision making as well. Many programs are being engineered to make predictions of the ESWL success rates based on the patients’ data. Some research studies applied a selected machine learning analysis to predict the result of the ESWL treatment. They examined the effect of SWL treatment by using machine learning methods and confirmed that prediction accuracy provides noteworthy results by using various patients and stone characteristics [37].

Moreover, some studies have shown that machine-learning algorithms, such as artificial neural networks, outperform clinicians’ image interpretation, and if applied, they could optimize the stone hit rate by up to 75% [33,38].

The importance of machine-learning algorithms can give matched insights to domain knowledge on effective and influential factors for SWL success outcomes. The new machine learning based on artificial intelligence and medical encounter represents an effective and modern approach for the coming years. When further large studies, validated in a prospective group of urinary stone patients, become available, the new machine learning methods might be useful for guiding SWL treatment selection and prediction of patients with urinary stones [26], and MLP could be the beginning of a new path for urology [37].

## 7. Conclusions

In conclusion, although ESWL requires expertise and nowadays is not considered a treatment of excellence, if we continue to see an evolution of newer models and this trend of artificial intelligence spiking, it could become a strong competitor of ureterorenoscopy once again. Even if this does not happen, extracorporeal shockwave lithotripsy will remain an excellent non-invasive option for kidney stone treatment for many years to come.

## Figures and Tables

**Figure 1 ijerph-20-04127-f001:**
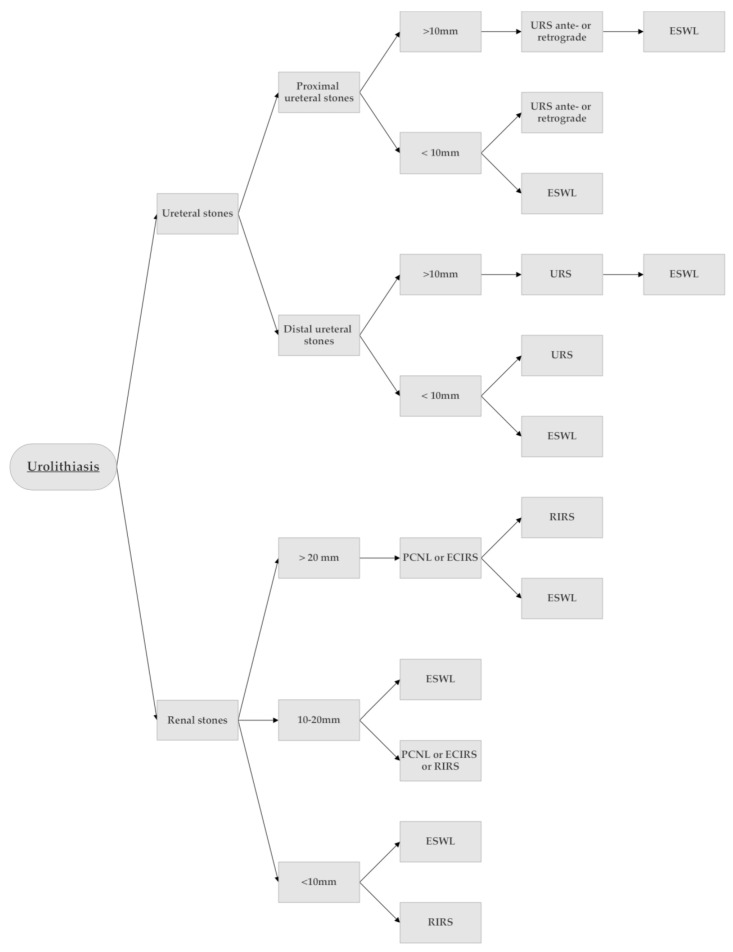
Treatment algorithm for renal and ureteral stones (inspired by EAU Guidelines. Edn. presented at the EAU Annual Congress Amsterdam, 2022 [1]).

**Figure 2 ijerph-20-04127-f002:**
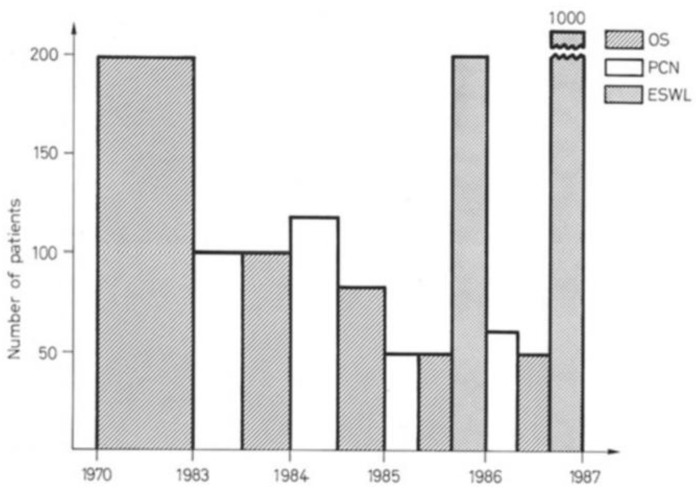
The diagram from the original article comparing surgical and non-surgical treatment for renal stones from 1970 to 1986 at our Department, University Hospital Genoa.

**Figure 3 ijerph-20-04127-f003:**
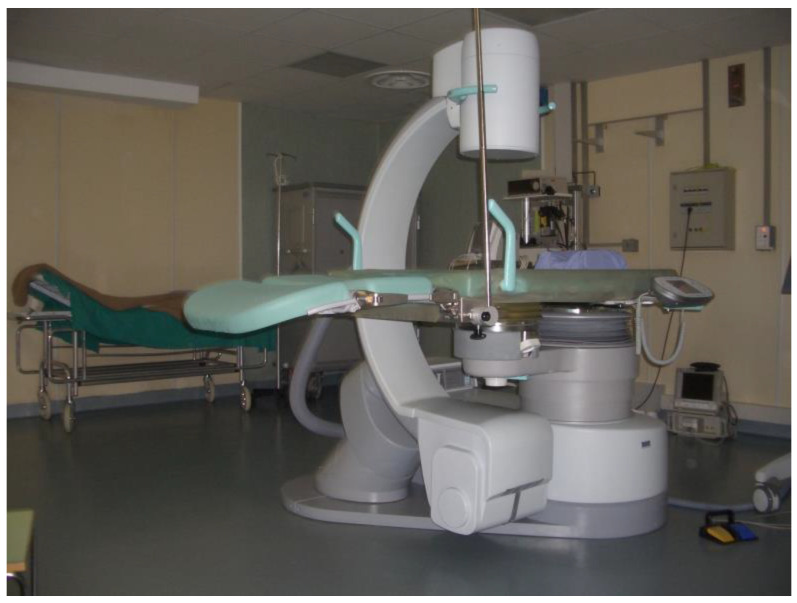
The Lithoring Multi-One lithotripter in its dedicated room at our ‘Kidney Stone Center’.

**Figure 4 ijerph-20-04127-f004:**
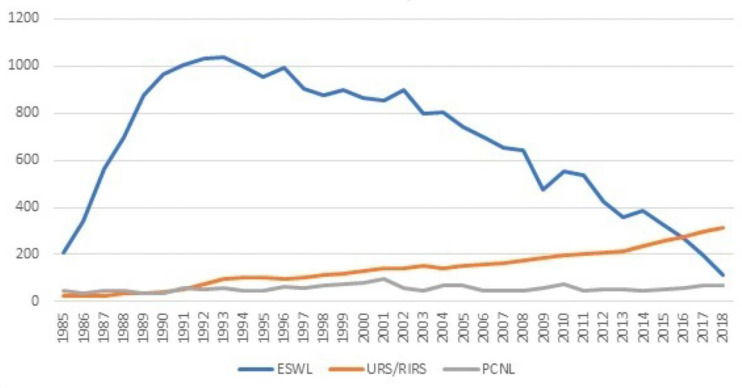
Kidney stone treatment modalities trends in our department from 1985 until 2018.

## Data Availability

Not applicable.

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
