# Peer review of "The Lifetime History of the First Italian Public Extra-Corporeal Shock Wave Lithotripsy (ESWL) Lithotripter as a Mirror of the Evolution of Endourology over the Last Decade"

_ijerph, 2023, doi:10.3390/ijerph20054127_

Round 1

Reviewer 1 Report

It is a good revision of the history of ESWL and the actual indication

In line 47: 12 mm or 12 cm?

About complication could describe the tipe in particular the major one like hematoma that required embolization or nephrectomy

Could you better explain why although ESWL is less expensive and less risky than RIRS, its use has greatly decreased?

In some line the number or references are before full stop

In some line more than one full stop

Author Response

February 20th, 2023

Prof. Dr. Paul B. Tchounwou

Editor in chief, IJERPH

We would like to thank you for reviewing our manuscript titled “The lifetime history of the first Italian public Extra-corporeal Shock Wave Lithotripsy (ESWL) lithotripter as a mirror of the evolution of endourology over the last decade.” and for the possibility of a new submission with the corrections suggested by your reviewer. We think our review is improved and complete and we hope now will be suitable for publication in the esteemed International Journal of Environmental Research and Public Health.

Kind regards

Dear reviewer, we would like to thank you for the comments.

We provide to correct each point you suggest to us:

“In line 47: 12 mm or 12 cm?”

  • We correct the sentence in “cm”, line 52.

“About complication could describe the type in particular the major one like hematoma that required embolization or nephrectomy”

  • We add a small paragraph, lines 55-58.

“Could you better explain why although ESWL is less expensive and less risky than RIRS, its use has greatly decreased?”

  • We add a sentence, lines 65-68.

“In some lines, the number of references is before full stop”

“In some line more than one full stop”

  • We provide to correct the punctuation.

We hope the new version of the manuscript will fulfill your expectations.

Best regards,

Guglielmo Mantica, MD

Reviewer 2 Report

I would like to congratulate the authors for their work on ESWL. It is true that current trend is to prefer URS/ RIRS/ PCNL/ ECRIS rather than ESWL and, for this reason, it is important that work on ESWL gets published in order to give further support to this technique. 

In my opinion, this article is suitable for publication, I would like to give one comment. In section 5, you could add a short paragraph regarding ECRIS, combination of URS flexible and PCNL for large renal calculi.

Best of luck! 

Author Response

February 20th, 2023

Prof. Dr. Paul B. Tchounwou

Editor in chief, IJERPH

We would like to thank you for reviewing our manuscript titled “The lifetime history of the first Italian public Extra-corporeal Shock Wave Lithotripsy (ESWL) lithotripter as a mirror of the evolution of endourology over the last decade.” and for the possibility of a new submission with the corrections suggested by your reviewer. We think our review is improved and complete and we hope now will be suitable for publication in the esteemed International Journal of Environmental Research and Public Health.

Kind regards

Dear reviewer,

we would like to thank you for the comments and for the appreciation of our manuscript.

We provide to correct each point you suggest to us:

“In section 5, you could add a short paragraph regarding ECIRS, combination of URS flexible and PCNL for large renal calculi.”

  • We add a small paragraph, lines 374-378.

We hope the new version of the manuscript will fulfill your expectations.

Best regards,

Guglielmo Mantica, MD

Reviewer 3 Report

This manuscript reviewed the history of ESWL in detail. However, endourology was another topics in managing the urinary tract stone. Could the author provide the comparison among ESWL, PCNL, RIRS, and ureteroscopy lithotripsy ? A Table could be utilized in the very beginning to make our readers to understand which choices are there for managing urinary tracts stones. 

Author Response

February 20th, 2023

Prof. Dr. Paul B. Tchounwou

Editor in chief, IJERPH

We would like to thank you for reviewing our manuscript titled “The lifetime history of the first Italian public Extra-corporeal Shock Wave Lithotripsy (ESWL) lithotripter as a mirror of the evolution of endourology over the last decade.” and for the possibility of a new submission with the corrections suggested by your reviewer. We think our review is improved and complete and we hope now will be suitable for publication in the esteemed International Journal of Environmental Research and Public Health.

Kind regards

Dear reviewer,

we would like to thank you for the comments and for the appreciation of our manuscript.

We provide to correct each point you suggest to us:

“Could the author provide the comparison among ESWL, PCNL, RIRS, and ureteroscopy lithotripsy ?”

  • We add an algorithm in the first section, as suggested, inspired by EAU Guidelines (Edn. presented at the EAU Annual Congress Amsterdam, 2022): figure 1.

We hope the new version of the manuscript will fulfill your expectations.

Best regards,

Guglielmo Mantica, MD

Reviewer 4 Report

It is a well documented paper but lacks the scientific importance

Minor spelling issues e.g "open surgery.Only two years after 183" - many cases where space would be necessary after period signs or others

Author Response

February 20th, 2023

Prof. Dr. Paul B. Tchounwou

Editor in chief, IJERPH

We would like to thank you for reviewing our manuscript titled “The lifetime history of the first Italian public Extra-corporeal Shock Wave Lithotripsy (ESWL) lithotripter as a mirror of the evolution of endourology over the last decade.” and for the possibility of a new submission with the corrections suggested by your reviewer. We think our review is improved and complete and we hope now will be suitable for publication in the esteemed International Journal of Environmental Research and Public Health.

Kind regard

Dear reviewer, we would like to thank you for the comments and for the appreciation of our manuscript.

We provide to correct what you suggest to us: 

“Minor spelling issues e.g "open surgery.Only two years after 183" - many cases where space would be necessary after period signs or others”

  • We provide to correct the grammar and punctuation.

We hope the new version of the manuscript will fulfill your expectations.

Best regards,

Guglielmo Mantica, MD